# Immunomodulatory Nanomedicine for the Treatment of Atherosclerosis

**DOI:** 10.3390/jcm10143185

**Published:** 2021-07-20

**Authors:** Linsey J. F. Peters, Alexander Jans, Matthias Bartneck, Emiel P. C. van der Vorst

**Affiliations:** 1Interdisciplinary Center for Clinical Research (IZKF), RWTH Aachen University, 52074 Aachen, Germany; lipeters@ukaachen.de; 2Institute for Molecular Cardiovascular Research (IMCAR), RWTH Aachen University, 52074 Aachen, Germany; 3Department of Pathology, Cardiovascular Research Institute Maastricht (CARIM), Maastricht University Medical Centre, 6229 ER Maastricht, The Netherlands; 4Department of Medicine III, University Hospital Aachen, 52074 Aachen, Germany; ajans@ukaachen.de (A.J.); mbartneck@ukaachen.de (M.B.); 5Institute for Cardiovascular Prevention (IPEK), Ludwig-Maximilians-University Munich, 80336 Munich, Germany; 6German Centre for Cardiovascular Research (DZHK), Partner Site Munich Heart Alliance, 80336 Munich, Germany

**Keywords:** cardiovascular disease, atherosclerosis, inflammation, nanomedicine, nanoparticles

## Abstract

Atherosclerosis is the main underlying cause of cardiovascular diseases (CVDs), which remain the number one contributor to mortality worldwide. Although current therapies can slow down disease progression, no treatment is available that can fully cure or reverse atherosclerosis. Nanomedicine, which is the application of nanotechnology in medicine, is an emerging field in the treatment of many pathologies, including CVDs. It enables the production of drugs that interact with cellular receptors, and allows for controlling cellular processes after entering these cells. Nanomedicine aims to repair, control and monitor biological and physiological systems via nanoparticles (NPs), which have been shown to be efficient drug carriers. In this review we will, after a general introduction, highlight the advantages and limitations of the use of such nano-based medicine, the potential applications and targeting strategies via NPs. For example, we will provide a detailed discussion on NPs that can target relevant cellular receptors, such as integrins, or cellular processes related to atherogenesis, such as vascular smooth muscle cell proliferation. Furthermore, we will underline the (ongoing) clinical trials focusing on NPs in CVDs, which might bring new insights into this research field.

## 1. Introduction

Cardiovascular diseases (CVDs) remain one of the leading causes of death worldwide, highlighted by the statement from the World Heart Federation that the number of deaths due to CVDs is at the moment around 17.3 million per year. It is anticipated that the number of deaths due to CVDs, in particular due to its main clinical outcomes myocardial infarction (MI) and stroke, will exceed 23.3 million by 2030 [1]. Besides this enormous impact on health, CVDs also have a high economic burden due to diagnostic and treatment costs, which are still rising every year. This underlines the importance of further research into and development of novel therapeutic approaches.

### 1.1. General Pathology of Cardiovascular Diseases

The main underlying cause of CVDs is atherosclerosis, a lipid-driven chronic inflammatory disease of the arterial wall [2]. The pathophysiological cascade is mainly initiated by hemodynamic shear stress leading to endothelial damage [3]. This results in increased endothelial permeability, consequently enabling the infiltration of various lipids such as low-density lipoprotein (LDL) into the intima. Here, LDL can be modified into oxidized LDL (oxLDL) by reactive oxygen species (ROS), which are released by the damaged endothelial cells (ECs) [4,5]. Both the accumulation of lipids and the endothelial damage trigger an inflammatory reaction, which includes the release of chemokines from the activated endothelium and the expression of adhesion molecules on ECs [4,5]. This inflammatory response stimulates the recruitment and adhesion of leukocytes, predominantly monocytes, to the endothelium [6]. Subsequently, these immune cells will transmigrate into the vessel wall, a process that is also guided by chemokines and adhesion molecules. The infiltrated monocytes will differentiate into macrophages, which take up oxLDL and cell debris, leading to the formation of foam cells [7,8]. Foam cells on their turn release inflammatory cytokines, which again triggers the recruitment cascade, thereby creating a vicious cycle. The continued recruitment of monocytes into the vessel wall results in the formation and further development of so called fatty-streak lesions. During lesion development, medial vascular smooth muscle cells (vSMCs) migrate towards the luminal side of the lesion where they produce collagen, which consequently leads to the formation of a fibrotic cap on top the macrophage rich areas of the plaque. In this stage, macrophages residing inside the lesion will not only secrete inflammatory cytokines, but also matrix metalloproteinases (MMPs), which break down extracellular matrix components in the plaque cap, thereby destabilizing the plaque. Moreover, due to continued lipid accumulation these macrophages become apoptotic, contributing to the formation of the necrotic core and thereby mediating plaque progression into an advanced stage [9]. Furthermore, in advanced human lesions angiogenesis plays an important role driven by macrophages [10,11]. Eventually, the continued growth of the plaque together with the thinning of the fibrotic cap can lead to plaque rupture and thrombus formation. The thrombus can then trigger other cardiovascular events, such as MI or stroke by occluding certain vessels in the heart or the brain, respectively [8].

### 1.2. Current CVD-Therapies

CVD management mainly revolves around the stabilization of blood lipid levels via statins and the reduction of thrombotic factors via for example aspirin, which result in slowing down disease progression. Statins are the current gold standard of CVD-therapy. They inhibit HMG-CoA reductase, consequently decreasing the production of cholesterol [12]. A meta-analysis on clinical statin trials unveiled that this medication can indeed reduce the LDL levels in plasma by 50–55% thereby also significantly decreasing the risk of further cardiovascular events [13]. However, treatment with statins also has a lot of off-target effects. For example, a study by Preiss et al. [14] revealed that treatment with statins increased the risk for the development of diabetes by a striking 9%. This discovery led to a discussion about the use of statins in the clinic and especially stimulated the development of alternative treatment options. One of these new strategies is the use of monoclonal antibodies against proprotein convertase subtilisin/kexin type 9 (PCSK9). Its physiological function is to stimulate the degradation of the LDL receptor via direct interaction in the liver [15]. Additionally, PCSK9 prevents LDL receptor recycling to the membrane. In this way, inhibiting PCSK9 will avert LDL receptor degradation and hence lead to increased surface expression of LDL receptors that can bind and internalize LDL particles, thereby lowering plasma LDL levels. Interestingly, PCSK9 inhibition can reduce plasma LDL levels by a striking 60%, even on top of statin-induced LDL decrease, without any signs of serious side effects [16,17,18]. Although PCSK9 inhibition is a very promising therapeutic option, the production costs of these antibodies remain at the moment too high in order for it to be used on a large scale.

Besides these lipid-focused CVD-therapies, novel methods based on immunomodulation have emerged in the last decades. The immune system protects the body from infections. In large parts, this is based on the recognition of “self” and “non-self”. Immunity is vital to protect the body, but exaggerated inflammation and elevated white blood cell counts are a risk factor for CVD [19] as described before. A study by Ridker and Luscher [20] showed the beneficial role of interleukin-1β (IL-1β), tumor necrosis factor (TNF) and interleukin-6 inhibition on CVD outcome, which are all a part of a common pathway. Initially, an inactive precursor of IL-1β is produced which requires proteolytic cleavage by the nucleotide-binding leucine-rich repeat-containing pyrin receptor 3 (NLRP3) inflammasome [21]. By using the monoclonal antibody canakinumab, IL-1β can be inhibited. Notably, this inhibition results additionally in a reduction of plasma IL-6 and high-sensitivity C-reactive protein (hsCRP) levels [22]. In the Canakinumab Anti-inflammatory Thrombosis Outcome Study (CANTOS) trial the effects of targeting IL-1β on the risk of cardiovascular events were evaluated. CANTOS was a double blind, randomized, placebo-controlled trial which involved stable patients with previous MI. This trial showed that inhibition of IL-1β by canakinumab was indeed effective in lowering plasma hsCRP levels and preventing further adverse cardiac events [23]. This study showed the great potential of the immunomodulatory drug canakinumab; however, the use of this drug was associated with increased fatal infections and sepsis. This occurred despite the exclusion of participants with chronic or recurrent infections. Additionally, targeting TNF has also been studied in several settings. While anti-TNF treatment has proven success in several inflammatory diseases such as rheumatoid arthritis, it has not been directly investigated in CVD patients [24]. However, 1 year of anti-TNF-α therapy showed beneficial effects on vascular function in rheumatoid arthritis and ankylosing spondylitis patients [25]. Although targeting of these cytokines seems a very promising therapeutic approach, there remains a high unmet need for interventions in CVD that aim at controlling the immune system.

## 2. Nanomedicine

As described above, CVD still puts the highest burden on both health and economy worldwide while there is still no treatment regimen to completely prevent or revert disease progression. A lot of the novel treatment options have either serious side effects, are too expensive to produce for large-scale use or have yet to be proven effective for CVDs. Recently, nanomedicine has gained interest as it could alleviate at least some of the disadvantages of the conventional medication.

### 2.1. What Is Nanomedicine?

Nanoscience is the science behind the production, manipulation, design and application of materials smaller than 1000 nm. It is a field that is gaining interest in many areas, such as biotechnology, medical technology, biology and medicine. In the field of medicine, nanotechnology has provided many innovative methods for diagnostics, imaging, analytics and treatment procedures, for example gene delivery systems, targeted drug delivery and tissue engineering scaffolds [26]. The application of nanotechnology in medicine is referred to as nanomedicine, which enables the generation of drugs that interact with cells on the level of their receptors, and to modify and direct cellular processes after entering these cells. Nanomedicine is devoted to repairing, controlling and monitoring biological and physiological systems via nanoparticles (NPs), which can act as carriers for drugs, proteins and nucleic acids [27]. While the definition includes particles sizing up to 999 nm, most therapeutic NPs are below 100 nm to avoid clearance by the cells by the reticuloendothelial system (RES), which includes ECs, monocytes and macrophages. However, large quantities of NPs are unavoidably internalized by the liver. Generally, one can distinguish inorganic and organic NPs. Inorganic NPs consist of metals, like silica, gold and silver, which allows imaging systems to trace them, while organic NPs are amongst others dendrimers, carbon nanotubes, liposomes and polymeric micelles [28,29]. Of the inorganic NPs, the gold NPs have been investigated the most. We have demonstrated that particularly macrophages can ingest large quantities of gold NPs [30]. Among the metallic nanoparticles, usage of iron oxide potentially enables an application of magnetic fields to guide the particles in the body [31]. However, the inorganic NPs suffer from the drawback of not being biodegradable, and thus organic NPs are suited best as carrier material for nanomedicine. Particularly lipid capsules have demonstrated to have a great usability due to their similarity with the natural cell membrane that is also composed of phospholipids [32,33]. Therefore, lipid-based drugs represent the most successful nanomedicine at the moment. However, it is important to realize that the usage of lipids for drug encapsulation strongly effects the pharmacokinetics of drugs. For instance, the chemotherapeutic Doxil, which is encapsulated doxorubicin, greatly enhances the circulation period of the drug [34]. Recently, lipid-based nanoparticles (LNPs) to encapsulate mRNA have also proven to be a huge success as a novel class of vaccines for severe acute respiratory syndrome coronavirus type 2 (SARS-CoV-2) [35]. Additionally, the clinically approved LNPs that are most famous at the moment are based on lipid encapsulation, i.e., the COVID-19 vaccines from BioNTech/Pfizer and Moderna, are lipid nanoparticles [35]. LNPs enter the cells by binding to the LDL-receptor aided by apolipoprotein E (ApoE), which stimulates its uptake significantly [36]. As the LDL-receptor is not only expressed on hepatocytes, but also on other cells like antigen-presenting cells, LNPs can target a variety of cell-types to exert its effects and modulate lipid and/or inflammatory mechanisms (Figure 1). Another lipoprotein that has been encapsulated into LNPs for receptor-mediated uptake is ApoA1, which has been extensively discussed in another review [37] and which will therefore only be briefly highlighted here. ApoA1 is the main protein that is present in high density lipoprotein (HDL) and is taken up in the liver via the scavenger receptor class B type I (SR-B1). Conveniently, NP uptake via SR-B1 can be achieved by engineering reconstituted HDL (rHDL) NPs containing various structures of ApoA1, which can alter the biological behavior of the NP [38]. One approach involves the use of ApoA1 mimetic peptides that contain amphipathic helix motifs comparable with those of ApoA1 or ApoE [39]. Some of these mimetic peptides have been shown to improve reverse cholesterol transport via ATP binding cassette transporter A1 (ABCA1) [40,41]. Besides targeting lipid metabolism, rHDL particles can also be used to influence immune check-points. For example, Seijkens et al. used rHDL NPs to deliver small molecule inhibitors blocking the interaction between tumor necrosis factor receptor-associated factor (TRAF) 6 and CD40 (TRAF-STOPs) to macrophages and hereby successfully treated atherosclerotic ApoE^−/−^ mice [42]. Lipoproteins and lipid nanoparticles show a high similarity regarding their physical, chemical and compositional properties. One main advantage of lipid-based nanoparticles is the storage ability, as LNPs can be stored in the range of 2 °C to −20 °C for more than 50 days and in lyophilized state even up to 11 months without the loss of gene silencing efficacy, compared to freshly prepared ones [43]. Depending on the physicochemical properties and surface modification, LNPs can remain more than 4 h in circulation in vivo before being cleared [44].

Overall, the use of lipid encapsulation for treatment strategies in various settings further demonstrates the potential of nanomedicine in the clinic. However, as briefly highlighted in the next section there are several advantages and limitations that should be kept in mind during the development and use of NPs.

### 2.2. Advantages and Limitations of Nanoparticles

In medicine, NPs have numerous advantages over traditional and modern therapies. These advantages make them ideally suited to be used in disease diagnosis and treatment. One technological advantage of NPs is their high stability, as it provides for a long shelf life. Additionally, NPs have a high carrier capacity, meaning that many drug molecules can be incorporated into the NP enabling the delivery of relatively high amounts of drug to the cells. Another advantage is the possibility to incorporate both hydrophilic and hydrophobic substances into the NP, making it a highly dynamic and flexible delivery method. Lastly, NP-based treatment has higher practicality in terms of administration as different routes, such as oral ingestion or inhalation, can be used to administer NP-based drugs [45]. Furthermore, NPs can be produced in a well-controlled manner regulating size and allowing further modifications which improve the overall drug delivery. In particular, size is a crucial factor that determines delivery efficiency. On the one hand small NPs will be excreted faster through the kidneys, while on the other hand larger particles are often trapped in the liver and spleen and therefore unable to reach their target cell/organ. One of the main routes for NP take-up is endocytosis, where the optimal size for NPs to induce potent endocytosis ranges from 30 to 60 nm. NPs smaller than 30 nm fail to induce endocytosis while the ones larger than 60 nm cause receptor overconsumption [46]. Furthermore, these particles can be processed in a way that allows additional conjugations of chemical molecules, peptides or electrical charges, which all improve targeting, water solubility, cellular uptake and structural stability [47,48]. For example, hybrid NPs that consist of two types of NPs, such as polymers and liposomes, further improve the benefits of each particle for drug delivery and imaging. Advantages of NPs do not only apply to drug delivery, but also to other medical treatment options. For instance, by embedding NPs into biomaterials a hybrid biomaterial system is created. Biomaterials like hydrogels are for example being used during surgery to stop bleeding and improve wound healing. A hydrogel can be combined with a 3D polymer structure network to better control the kinetics of both the NPs itself that are released from the hydrogel and of the drugs released from the NPs. Additionally, NP clearance is accelerated by the biodegradable property of the hydrogel for detoxification [49]. 

Besides the many advantages of the use of NPs, there are also some downsides. One of the downsides is the potential “nano-toxicity” of these particles. NPs could accumulate in the blood and organs which could be harmful. For example, a study by Miller et al. showed that gold NPs were preserved in the urine and blood of healthy subjects for three months after administration via inhalation [50]. Although the gold particles did not show an immediate adverse effect on health, the accumulation of these particles in organs could be problematic for long term treatments. Moreover, NPs could potentially be contaminated with small amounts of bacterial lipopolysaccharide (LPS) or endotoxin during the particle preparation. Consequently, the LPS could induce inflammasome activation in immune cells by binding to receptors expressed in these cells such as Toll-like receptor 4 (TLR4) [51]. However, contamination of the NPs is extremely sparse when working carefully so the chance that these off-target effects occur is very low.

Overall, the benefits of NPs seem to outweigh their disadvantages and NPs definitely have several advantages compared to traditional and other modern medical strategies. To overcome any safety issues for NP applications, NPs must be designed carefully followed by a series of experimental tests to confirm their functionality. As atherosclerosis is an inflammatory disease, the reaction of immune cells to NPs and vice versa should also be taken into consideration in their design, as discussed below.

### 2.3. Nanoparticles and Their Interactions with Immune Cells

As described before, the immune system functions mostly by recognition of the body’s own particles and foreign bodies. Therapeutic NPs will in any case have unintended interactions with the RES as they can be recognized by the immune system as foreign [52]. Several immune cells, such as monocytes, macrophages, dendritic cells (DCs) and polymorphonuclear leukocytes will, upon recognition, phagocytose these NPs for clearance. Once degraded, NPs have been shown to be able to reprogram macrophage polarization by inducing or suppressing gene expression [53,54]. Furthermore, these particles stimulate inflammatory cytokine production by immune cells and elicit complement activation in the blood [55,56,57,58]. In contrast, exposure to silver or cationic lipid NPs could lead to neutrophil elimination as it triggers neutrophil extracellular trap formation, release and cell death [59,60]. Thus, NPs can trigger various immune responses on their own, independent from whether they carry drugs or not. There have been various strategies to avoid interactions of NPs with the RES. For example, reduction of unspecific binding to cells and proteins is achieved by PEGylation, which is the process of attaching strands of the polymer polyethylene glycol (PEG) [61]. Although this method is used frequently, many researchers oversee the fact that PEG will accumulate in the body and that an immune response in the form of antibody production takes place against PEG. This is most likely responsible for the more rapid clearance of liposomes from the blood upon repeated NP administration [62]. Another method to reduce unintended interactions with the RES is the use of coatings that decrease the acute immune reaction, such as poly(N-isopropylacrylamide) hydrogel particles crosslinked with PEG diacrylate. In this way the immune response is optimized, avoiding unwanted triggering of the RES [63]. With regard to leukocytes, it is known that monocytes, macrophages and DCs recognize pathogens, internalize them and present the antigens of novel pathogens via the major histocompatibility complex II (MHC-II) to T helper cells [64]. Subsequently, by secretion of cytokines the T helper cells then recruit other immune cells such as cytotoxic T cells and B cells, which kill infected cells and produce antibodies against antigens, respectively. It has been shown that surface chemistry, material composition and peptide modifications of NPs can affect DC maturation, thereby possibly also affecting T cell instruction by DCs [64]. Besides their direct effect on monocytes, macrophages and DCs, NPs could also interact with the MHC-II complex on these cells to influence the immune response as shown by Clemente-Casares et al. [65]. It is important to note that several studies have highlighted that there is a discrepancy in NP uptake comparing in vitro and in vivo experiments. For example, Bartneck et al. [64] showed that when ECs were cultured as a monolayer they took up large amounts of NPs, while another study showed that in vivo the ECs did not internalize NPs in the liver, as these particles were only found in hepatic macrophages [30]. This highlights the need for further in vivo studies to evaluate the potential of nanomedicine-based strategies for immunomodulation as a potential CVD-therapy. The interactions of different LNP formulations with leukocytes in the context of liver fibrosis were recently elaborately reviewed elsewhere [66].

## 3. Therapeutic Nanomedicine-Based Approaches

As mentioned before, current CVD-therapies are far from perfect. Nanomedicine can provide interesting, and perhaps better, alternatives to the traditional lipid- or immunomodulatory drugs. Nanotechnological drugs could work especially well for the treatment of atherosclerosis as NPs are being phagocytosed by macrophages, the most abundant cell type in atherosclerotic plaques. This, together with their phagocytotic capacity opens up a new road to deliver drugs to macrophages inside atherosclerotic plaques [67]. In the following sections we will discuss potential approaches to treat CVDs with nanomedicine.

### 3.1. Targeting Cellular Receptors

One can distinguish two main levels on which leukocytes can be targeted with drugs: the first can occur on the level of their receptors, and the second can be exerted from modifying biological processes inside the white blood cells. The first option can aim at leukocyte receptors that guide them through the vascular wall: the circulating leukocytes are “called in” by any injured organ through the secretion of specific cytokines or chemokines which attract the leukocytes. The process of endothelial transmigration is based on the expression of various adhesion molecules such as selectins and integrins [68]. Integrins are membrane glycoproteins that can bind to various components in the extracellular matrix surrounding them. Blocking selectins and integrins was shown to be a successful strategy in limiting the infiltration of additional immune cells which otherwise would further amplify inflammation and lead to organ damage. Besides transmigration, the integrins like ανβ3 integrins also play roles in other atherosclerosis-related processes, such as angiogenesis. Integrin ανβ3 is expressed on ECs and its expression is upregulated by pro-angiogenic factors [69]. This integrin can then be used to deliver NPs into atherosclerotic plaques by coating the NPs with cRGDfK peptides. These cRGDfK peptides specifically bind to ανβ3, thereby allowing for entrance into the cell [70]. Another receptor that can be targeted with NPs is stabilin-2 (STAB2). This is a transmembrane receptor that is involved in angiogenesis and cell adhesion. STAB2 can bind and clear multiple ligands such as heparin and hyaluronic acid (HA) [71]. In atherosclerotic plaques, macrophages and ECs have abundant surface expression of STAB2 [72]. HA, which is highly biocompatible and biodegradable, was incorporated into NPs and used to visualize atherosclerotic plaques by binding to macrophages and diseased endothelium [73]. HA is also a ligand for other receptors such as CD44 [74]. This receptor is involved in the activation of immune and vascular cells and in cell adhesion of leukocytes to the endothelium and is known to be pro-atherogenic by stimulating the production of pro-inflammatory mediators [75,76]. Furthermore, inhibition of the receptor led to reduced leukocyte recruitment and vSMC activation [77]. Thus, targeting this receptor with HA NPs could be an interesting strategy to target atherosclerotic lesions and therefore usable for the treatment of CVDs.

### 3.2. Targeting Cellular Processes

The second level of immune cell manipulation can be considered by intervening with cellular processes. One of the key processes in atherosclerosis is vSMC proliferation and death, which ultimately leads to the development of the necrotic core, thinning of the fibrotic cap and calcification [78]. To target these processes, various caspase inhibitors have been tested such as rapamycin. Rapamycin was for example loaded into gel-like NPs. These NPs showed a 20% reduction in human vSMC proliferation in vitro compared to regular, free rapamycin [79]. The same study also tested these NPs in rats in vivo by infusion into the left common carotid artery after vascular injury. Interestingly, the concentration of rapamycin in the carotid artery remained stable for 2 weeks and was not detectable in the opposite carotid, indicating a localized delivery. The treated rats showed a significant decrease of hyperplasia and increased reendothelialization of the injured artery. Overall, this study thus shows the potential of NPs in treating CVDs by targeting cellular processes. Another important process in the pathophysiology of CVDs is inflammation. By targeting the process of inflammation, other processes initiated by the increased inflammatory state could also be reduced indirectly. For example, Meneghini et al. injected chemotherapy drug docetaxel (DTX)–loaded NPs that are lipid-based (LDE) intravenously into a rabbit model of atherosclerosis [80]. Notably, the levels of pro-inflammatory markers NF-κB, IL-6, IL-1β and TNF-α were all significantly decreased by LDE-DTX. Possibly as a result of the decreased inflammatory state, treatment with LDE-DTX resulted in a striking 80% reduction of atherosclerotic plaque area compared to controls. Additionally, expression of chemokine (C-C motif) ligand 2 (CCL2), which is involved in monocyte infiltration, and the macrophage marker CD68 were significantly lowered by LDE-DTX treatment, strongly suggesting that leukocyte infiltration was inhibited, possibly via the lowered inflammatory state. We have demonstrated before that encapsulated dexamethasone is applicable to treat liver diseases, both in acute and chronic settings [33]. More specifically, targeting immune cells with small interfering RNA (siRNA) for transcription factors was demonstrated to be efficient to circumvent inflammatory activation [81]. Additionally, a recent study by Tao et al. [82] showed that the Ca2+/calmodulin-dependent protein kinase γ (CaMKIIγ) gene can be targeted with siRNA NPs in Ldlr^−/−^ mice. CaMKIIγ is activated in plaque macrophages and stimulates necrosis, which is also a key cellular process in atherosclerosis development. Inhibition of this gene via siRNA NPs indeed resulted in decreased necrotic area in the plaques of these mice and an overall increased plaque stability. Combined, these results clearly demonstrate the great potential that NPs have to target cellular processes in light of atherosclerosis and CVD.

### 3.3. Targeting Lipid Levels

Besides the targeting of cellular receptors and processes, one can also target other key characteristics of CVD such as the increased LDL plasma levels. As described before, a popular target for the reduction of plasma LDL is PCSK9. In the more classical strategies, PCSK9 is inhibited via monoclonal antibodies leading to increased LDL receptor expression and consequently lower levels of LDL in the plasma. Another way of interfering with this process is by inhibiting the synthesis of PCSK9 via siRNAs. As their stability is limited in vivo, a nanocarrier is required for the cellular delivery of siRNAs. In a study by Fitzgerald et al. [83], the efficacy of ALN-PCS, a PCSK9 expression inhibitor, was tested in healthy volunteers with increased cholesterol level who were not using any lipid-lowering drugs. ALN-PCS is a siRNA that inhibits PCSK9 synthesis and was delivered through lipid NPs. The treatment-emergent adverse events in the group of volunteers who received ALN-PCS were not different compared to the group who received the placebo, strongly suggesting that ALN-PCS is safe to use. Interestingly, ALN-PCS treatment reduced PCSK9 levels by 70% and LDL cholesterol levels by a striking 40% compared to placebo treatment. Overall, this study shows the great potential of nano-engineered drugs that target lipid levels to treat CVDs. Although, it should be kept in mind that these effects have been observed in healthy volunteers and therefore the additive effects on top of for example statin treatment remains to be determined.

Interestingly, a nanomedical drug was approved in December 2020 in the EU for use in adults with primary hypercholesterolemia (heterozygous familial and non-familial) or mixed dyslipidemia, as an adjunct to diet: Inclisiran (Leqvio^®^; Novartis, Basel, Switserland) is a first-in-class, siRNA conjugated to triantennary N-acetylgalactosamine carbohydrates (GalNAc). Inclisiran targets hepatocytes in the liver by binding to the asialoglycoprotein receptor aided by p-aminophenyl δ-D-galactopyranoside (Figure 1). It functions, similar to the statins, by lowering cholesterol levels. The advantage is that it is administered subcutaneously and only twice yearly [84].

LDL itself is also a promising candidate for NP-based medicine as it accumulates inside atherosclerotic plaques. Sobot et al. suggested to target LDL-accumulating cancer cells by using the cholesterol pre-cursor squalene conjugated with drugs [85,86]. This strategy could also be applied to target LDL-accumulating macrophages in atherosclerotic plaques, but for now only imaging methods using florescent markers have been established [87]. Although further research is necessary in this area, targeting LDL with NPs seems to be a promising strategy to treat atherosclerosis and further CVDs.

## 4. Clinical Trials

One of the promising aspects of nanomedicine is the possibility to revive the clinical potential of certain therapeutics via their reformulation. An example of this is the nanomodulation of the abandoned drug wortmannin (Wtmn), an inhibitor of phosphatidylinositol 3′ kinases (PI3Ks) and phosphatidylinositol 3′ kinase-related kinases (PIKKs). The drug was shown to be an effective therapeutic in the pre-clinical stages, but failed to be translated into the clinic because of low stability, poor solubility and high toxicity. A polymeric NP formulation of Wtmn was shown to overcome these negative aspects [88], thus proving the additional value of nanomedicine for translating medicine successfully into the clinic. In order to use NPs in the clinic, their safety and optimal dosage needs to be tested in clinical trials first. Besides the clinical study by Fitzgerald et al. [83] on the effects of ALN-PCS, other clinical trials have been performed testing drugs based on nanomedicine. Table 1 shows an overview of the completed clinical trials that include nano-engineered drugs for the treatment of CVDs. Although many clinical trials involving nanomedicine have been performed in various fields, only few of them specifically aim to target CVDs. However, these trials show promising results which highlight exactly how nanomedicine can improve current treatments. One of these studies is for example the clinical trial by Van der Valk et al. [89] in which liposomal encapsulation of the drug prednisolone phosphate improved its half-life in vivo. Unfortunately, no favorable effects of the drug were seen regarding inflammation, but the study still shows the benefits of adding nanotechnology to existing medicines. Another promising clinical trial focused on liposomal prostaglandin E_1_ (lipo-PGE_1_) as additional therapy to surgery in patients with acute lower limb ischemia (ALLI) [90]. The therapy was shown to be beneficial as the overall incidence of adverse clinical events was significantly lower in patients receiving lipo-PGE_1_ as compared to the control group (8.2% vs. 20.8%, respectively) [90]. Moreover, in a phase I clinical trial by Margolis et al. [91], the drug paclitaxel was bound to albumin and intravenously injected into patients with multivessel disease. In this manner the safety and optimal dosage was determined and results suggested that dosages below 70 mg/m^2^ were optimal in these patients. Table 2 and Table 3 provide an overview of the clinical trials that are still active or have been terminated, respectively. The ongoing clinical trials (Table 2) are in more advanced stages (i.e., phase II/III) than the completed trials (Table 1), thereby hopefully bringing nanomedicine one step closer to being used in actual clinical settings. An important fact to stress is that the terminated trials (Table 3) have been terminated based on reasons other than safety concerns, further stressing the safety and potential of nanomedicine as clinical approach.

## 5. Conclusions and Future Perspectives

To summarize, CVD remains a major contributor to both the global economic and health burden. Many diagnostic and treatment options are available, but these unfortunately still have some major drawbacks. Nanomedicine has the potential to overcome some of the hurdles that conventional medicine faces. Some of these advantages lie in their high stability, high carrier capacity, and the various ways of administrating nano-engineered drugs. One of the few limitations of the use of NPs is their potential toxicity. However, as proven by the various clinical trials performed with NPs, the use of these type of drugs is considered rather safe. Some issues regarding their application in CVD also remain for now. A key issue in targeting the vessels with nanomedicine will probably always be the superior clearance by the liver. However, when accessing monocytes in the blood stream as precursors for macrophages, or of neutrophils that might serve as Trojan horses, and further directing these cells into the vascular environment might help to inhibit the development of atherosclerotic lesions. Furthermore, the leukocyte recruitment cascades and the vascular and leukocyte receptors involved in this process are a potential interesting point of intervention using nanomedicines that should be investigated in greater detail in the future (Figure 2).

Overall, nanomedicine can be applied in various ways to aid current treatment options and further exploration of the use of NPs in CVDs will pave the way for nanomedicine in reducing the global burden of CVDs.

## Figures and Tables

**Figure 1 jcm-10-03185-f001:**
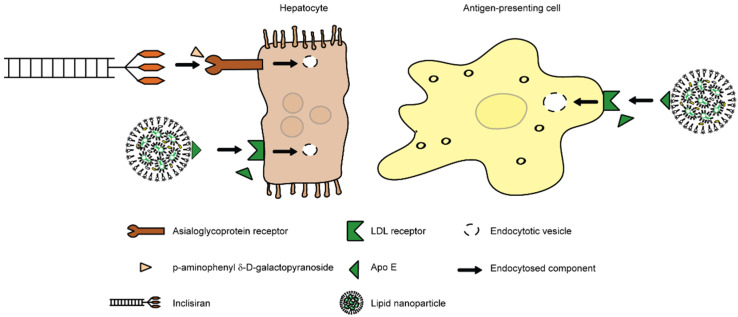
Mechanisms of cell infiltration by Inclisiran and LNPs. The figure demonstrates the different routes of cellular infiltration of two clinically used nanomedicines, being Inclisiran targeting hepatocytes and lipid nanoparticles (LNPs) targeting hepatocytes and antigen-presenting cells.

**Figure 2 jcm-10-03185-f002:**
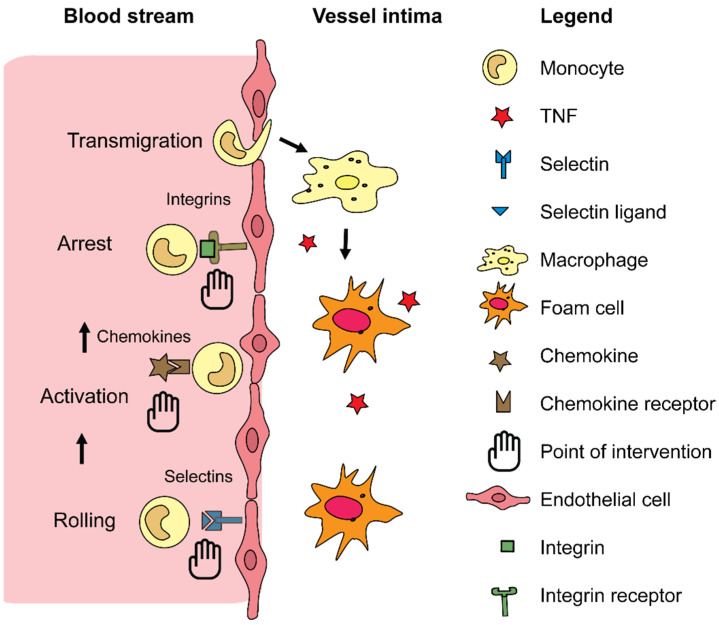
Future potential points of intervention for nanomedicine. The figure shows a schematic representation of leukocyte recruitment into the vessel wall and subsequent macrophage differentiation, the main process that drives atherosclerosis. Interesting potential points of intervention are highlighted that in the future might be promising processes at which nanomedicine can intervene as therapeutic approach.

**Table 1 jcm-10-03185-t001:** Overview of completed clinical trials which include nanomedicine for CVDs.

Nanocarrier	Drug	Phase	Effects/Outcome	Reference
Liposomal	ALN-PCS	Phase I	70% decrease in PCSK9 levels and 40% reduction of LDL cholesterol.	[83]
Liposomal	Prednisolone phosphate	Phase I/II	Liposomal encapsulation improved the half-life of the drug and successful delivery to plaque macrophages was achieved. However, no anti-inflammatory effects have been observed.	[89]
Liposomal	Prostaglandin E1 (PGE1)	Not specified	Patients with acute lower limb ischemia (ALLI) that received PGE1 had significantly less adverse events.	[90]
Albumin-bound	Abraxane/Paclitaxel	Phase I	The safety and optimal dose of Paclitaxel was tested in patients with multivessel disease. Dosages below 70 mg/m^2^ were tolerated by the patients and no adverse events were noted.	[91]
Silica gold and silica gold iron-bearing	-	Not applicable	Patients with atherosclerotic lesions received either silica gold NPs in an on-artery patch (Nano); silica gold iron-bearing NPs with targeted microbubbles and stem cells via a magnetic navigation system (Ferro); or a stent (Control). In both experimental groups the total atheroma volume was reduced up to 60 mm^3^ with a high level of safety. A five-year follow-up showed a higher safety and better mortality rate in the Nano group compared to the Ferro and Control.	[92,93]
Iron oxide-bearing	-	Not applicable	After acute MI, NPs of iron oxide were injected intravenously into patients and detected via magnetic resonance imaging. These NPs were taken up in the infarcted and remote myocardium, thus highlighting the method’s potential to be used to assess cellular myocardial inflammation and left ventricular remodeling.	[94]

**Table 2 jcm-10-03185-t002:** Overview of ongoing clinical trials which include nanomedicine for CVDs.

Nanocarrier	Drug	Phase	Aim Study	ClinicalTrials.Gov Identifier
Carbonaceous	-	Not applicable	The aim is to uncover whether eicosanoids drive the cardio-vascular effects of inhaled NPs. This will aid the design of NPs and circumvent the harmful characteristics of particles found in air pollution.	NCT03659864
Cholesterol-rich non-protein (LDE)	Methotrexate (MTX)-LDE	Phase II/III	The tolerance of MTX-LDE and the beneficial effects on plaque size will be tested in patients with aortic and coronary atherosclerotic disease.	NCT04616872
LDE	Paclitaxel-LDE	Phase II/III	The aim is to test the drug tolerance and effect on plaque size in patients with aortic and coronary atherosclerotic disease.	NCT04148833

**Table 3 jcm-10-03185-t003:** Overview of terminated clinical trials which include nanomedicine for CVDs.

Nanocarrier	Drug	Phase	Aim Study	Reason Termination	ClinicalTrials.Gov Identifier
Albumin-bound	Abraxane/Paclitaxel	Phase II	The preventative effects of the drug were studied on restenosis following revascularization of the superficial femoral artery (SFA).	Not based on the safety of the drugs or outcome of study, but terminated due to changing sponsor priorities.	NCT00518284
Silica gold with iron oxide	-	Phase I	Study whether a multistep approach, including the injection of the NPs, would achieve better results than conventional stenting.	Got terminated under pressure of the Federal Security Service of the Russian Federation.	NCT01436123
Iron oxide-labeled	-	Not applicable	Label blood cells from patients who had recent MI, reinject the NP-labeled cells and track the fate of these cells over the course of months. This would allow for better understanding of the role of inflammatory cells in MI and the technique could help the improvement of current treatments.	Not specified	NCT01127113

## Data Availability

Not applicable.

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
