# Peer review of "Immunomodulatory Nanomedicine for the Treatment of Atherosclerosis"

_jcm, 2021, doi:10.3390/jcm10143185_

Round 1
Reviewer 1 Report
Dear Authors
I thought that this review manuscript is interesting for the readers.
I have three comments.
- I read this manuscript, and thought this is interesting. I firstly read the abstract, but it had difficult that I imagined a summary. Authors will correct the abstract, for example added the content about nanomedicine of siRNAs which inhibits PCSK9 synthesis.
- This manuscript contained some drug name, DTA etc.. Authors should add the explanation carefully.
- Authors should add the journal name, years, no, and page in the reference section.
Sincerely yours.
Author Response
Dear Authors
I thought that this review manuscript is interesting for the readers.
I have three comments.
- I read this manuscript, and thought this is interesting. I firstly read the abstract, but it had difficult that I imagined a summary. Authors will correct the abstract, for example added the content about nanomedicine of siRNAs which inhibits PCSK9 synthesis.
Thank you for your kind suggestions. We have added some more specific examples of the content of our review in the abstract and believe this will indeed summarize the review better (Lines 26-29 of revised manuscript).
- This manuscript contained some drug name, DTA etc.. Authors should add the explanation carefully.
We understand the reviewer’s point and have added a brief description of the drugs such as DTX and ALN-PCS (Lines 337-338 and 365 of revised manuscript).
- Authors should add the journal name, years, no, and page in the reference section.
We fully agree with the reviewer and have made the required adjustments in the reference section.
Reviewer 2 Report
The authors have presented an informative review on the uses of nanoparticles for therapeutic treatment of atherosclerosis.
A few recommendation to the authors for inclusion in their review:
1) It was mentioned that ApoE is utilized in the Pfizer/Biontech Covid mRNA vaccine for targeted uptake via LDLR. What other sort lipoproteins or targeting agents have been embedded into nano-particles for drug delivery and receptor mediated uptake? Mention and discuss.
2) Are there any reports on incorporation of apoA-1 into a synthesized lipid nanoparticle in the shape of nascent HDL or spherical HDL for improved reverse cholesterol transport? Mention and discuss.
3) Please discuss the stability profile of some lipid nanoparticles in circulation for drug delivery and outside of the body. Are they similar to lipoprotein particles stability wise?
4) Are there any studies looking at vascular repair or treatment of chronic atherosclerosis using inhaled lipid nanoparticles containing drugs. Mention and discuss.
Author Response
The authors have presented an informative review on the uses of nanoparticles for therapeutic treatment of atherosclerosis.
A few recommendation to the authors for inclusion in their review:
1) It was mentioned that ApoE is utilized in the Pfizer/Biontech Covid mRNA vaccine for targeted uptake via LDLR. What other sort lipoproteins or targeting agents have been embedded into nano-particles for drug delivery and receptor mediated uptake? Mention and discuss.
2) Are there any reports on incorporation of apoA-1 into a synthesized lipid nanoparticle in the shape of nascent HDL or spherical HDL for improved reverse cholesterol transport? Mention and discuss.
Thank you for your recommendations. Since the first and second point overlap, we will respond here to both suggestions. We agree with the reviewer that these are interesting points to discuss and have therefore written a short paragraph about reconstituted HDL/ApoA1 and their potential as nanoparticles (Lines 165-180 of revised manuscript).
3) Please discuss the stability profile of some lipid nanoparticles in circulation for drug delivery and outside of the body. Are they similar to lipoprotein particles stability wise?
This is indeed an important aspect that we have now incorporated in our manuscript (Lines 180-186 of revised manuscript).
4) Are there any studies looking at vascular repair or treatment of chronic atherosclerosis using inhaled lipid nanoparticles containing drugs. Mention and discuss.
Although this is a very interesting point, as far as we know and can find there is currently no literature available yet regarding this aspect in light of inhaled nanoparticles, which therefore remains an exciting focus for future research.